# Diagnostic Value of the Neutrophil-to-Lymphocyte Ratio (NLR) and Platelet-to-Lymphocyte Ratio (PLR) in Various Respiratory Diseases: A Retrospective Analysis

**DOI:** 10.3390/diagnostics12010081

**Published:** 2021-12-30

**Authors:** Milena-Adina Man, Lavinia Davidescu, Nicoleta-Stefania Motoc, Ruxandra-Mioara Rajnoveanu, Cosmina-Ioana Bondor, Carmen-Monica Pop, Claudia Toma

**Affiliations:** 1Department of Medical Sciences, Pulmonology, “Iuliu Hatieganu” University of Medicine and Pharmacy, 400000 Cluj Napoca, Romania; manmilena50@yahoo.com (M.-A.M.); andra_redro@yahoo.com (R.-M.R.); cpop@umfcluj.ro (C.-M.P.); 2Faculty of Medicine and Pharmacy, University of Oradea, 410087 Oradea, Romania; lavinia.davidescu@yahoo.com; 3Department of Medical Biostatistics, “Iuliu Hatieganu” University of Medicine and Pharmacy, 400000 Cluj Napoca, Romania; cosmina_ioana@yahoo.com; 4Faculty of Medicine, Carol Davila University of Medicine and Pharmacy, 020021 Bucharest, Romania; claudia.toma@yahoo.co.uk

**Keywords:** respiratory diseases, inflammatory markers, neutrophil-to-lymphocyte ratio (NLR), platelet-to-lymphocyte (PLR)

## Abstract

The neutrophil-to-lymphocyte ratio (NLR) and platelet-to-lymphocyte (PLR) ratio are two extensively used inflammatory markers that have been proved very useful in evaluating inflammation in several diseases. The present article aimed to investigate if they have any value in distinguishing among various respiratory disorders. One hundred and forty-five patients with coronavirus disease 2019 (COVID-19), 219 patients with different chronic respiratory diseases (interstitial lung disease, obstructive sleep apnea(OSA)-chronic obstructive pulmonary disease (COPD) overlap syndrome, bronchiectasis) and 161 healthy individuals as a control group were included in the study. While neither NLR nor PLR had any power in differentiating between various diseases, PLR was found to be significant but poor as a diagnostic test when the control group was compared with the OSA-COPD group. NLR was found to be significant but poor as a diagnostic test when we compared the control group with all three groups (separately): the OSA-COPD group; interstitial lung disease group, and bronchiectasis group. NLR and PLR had poor power to discriminate between various respiratory diseases and cannot be used in making the differential diagnosis.

## 1. Introduction

The neutrophil-to-lymphocyte ratio (NLR) and platelet-to-lymphocyte ratio (PLR) are systemic inflammatory responses markers. Inflammation induces an increase in neutrophils, and platelet count accompanied by a decrease in lymphocyte count, making their ratios a valuable tool in indirectly evaluating both inflammatory status as well as cell-mediated immunity [1]. Platelets have a crucial role in the immune system due to the surface receptors that recognize pathogens and immune complexes. Activated and adherent platelets release cytokines, including chemokines that stimulate the cells’ recruitment [1,2]. NLR and PLR, together or separately, have been evaluated in several conditions such as malignancies (hematological malignancies included), respiratory diseases, gastrointestinal and cardiovascular (acute coronary syndrome, intracerebral hemorrhage), systemic diseases, and lately, coronavirus disease 2019 (COVID-19) [3]. Higher values have been associated with more severe forms of the disease and worse prognosis [1,2,3,4,5,6,7,8,9,10]. Higher values have also been recorded in acute versus chronic conditions [6,8], and the most elevated values have been reported in the presence of bacteremia [7]. As far as respiratory diseases go, these markers have been studied in stable chronic obstructive pulmonary disease (COPD), acute exacerbation of COPD, sleep apnea, bronchiectasis, interstitial lung diseases, and in the last two years in COVID-19 [3,4,5,6,7,8,9,10]. Although they have been assessed in several respiratory disorders, few studies have evaluated their diagnostic value among different respiratory diseases. Additionally, a cutoff value is still to be found. Therefore, this study aimed to assess the diagnostic value of NLR and PLR in various respiratory diseases and, if possible, to find a cutoff value for each respiratory disease.

## 2. Materials and Methods

### 2.1. Study Design

The present paper was a phase II retrospective diagnostic test studyof a group of several patient databases from our hospital, a clinical teaching hospital in one of the central cities of Romania, considered as cases. All patients that had neutrophil, lymphocyte, and platelet data in their files (both neutrophil-to-lymphocyte ratio and platelet-to-lymphocyte ratio could be calculated) were included in the study. Some patients were included in other studies [3,8,9,10,11]. In total, 364 patients with various respiratory diseases (145 with COVID-19-data for the entire group were published before in [3]), 36 patients with interstitial lung diseases, 98 patients with bronchiectasis, and 85 with obstructive sleep apnea (OSA)–chronic obstructive pulmonary disease (COPD) overlap syndrome) and 161 healthy individuals as the study control group were included. The control group included healthy medical staff that presented for annual evaluation. They were clinically examined and had blood tests performed. The patients were evaluated before the COVID-19 pandemic started. Data were collected between 2017–2021.

COVID-19 diagnostics were confirmed using real-time reverse-transcriptase polymerase-chain-reaction (RT-PCR) assay to test nasal and pharyngeal swab specimens according to World Health Organization (WHO)guidance. Patients’ characteristics and description data were already published, and they will not be referred to in the present paper [3]. Interstitial lung disease was diagnosed according to international criteria and after a multidisciplinary meeting [9], bronchiectasis was confirmed by a chest computer scan (chest-CT) [3], and OSA–COPD overlap syndrome was defined as the presence of OSA (positive ventilatory polygraphy after the recommendations of the American Sleep Society 2007) and COPD (GOLD 2020) in the same patient [10,11]. We excluded patients with other diseases that could cause high NLR and PLR values: patients with any type of cancer, hematological diseases, severe cardiac disease (NYHA III and IV cardiac failure, recent myocardial infarction in last 3 months, unstable arrhythmia), liver disease and systemic diseases. The control group was clinically examined by a full-fledged physician and had blood tests performed. All blood tests were performed in the hospital laboratory with standard procedures. The NLR ratio was defined as the absolute count of neutrophils divided by the absolute count of lymphocytes. The PLR was defined as the absolute count of platelets divided by the absolute count of lymphocytes. CRP (C-reactive protein) and ESR (erythrocyte sedimentation rate) were also determined.

The study protocol was reviewed and approved by the Ethics Committee for Scientific Research of the Hospital 232, approved on 03 March 2020.

### 2.2. Statistical Method

Statistical analysis was performed using IBM SPSS STATISTICS 25.0 application. Medians (25th percentiles; 75th percentiles) were calculated for quantitative variables with a non-normal distribution. Normal distribution was tested with the Shapiro–Wilk test. The comparison of multiple means was performed using an Anova test for independent samples with equal variations depending on the Levene test result. In cases where equal variations were not found, the Kruskal–Wallis test was used. Post-hoc analysis was performed to correct type I error with Sheffe’s method respective to the Bonferroni method. Frequencies were compared with the Chi-square test. For the best cutoff to discriminate between two groups in the case of a quantitative variable, the ROC (receiver operating characteristics) curve analysis was used. The area under the ROC curve (AUC) was reported. AUC was considered statistically significant when compared with 50% of the square of the area, in which case it meant that the considered parameter had the power to discriminate between the tested groups. The higher the AUC, the better the discrimination parameter. The optimal cutoff was considered when the Youden index was maximized, i.e., sensitivity (Se) plus specificity (Sp) minus 1. To evaluate the accuracy of the diagnostic test, the traditional academic point system was considered: 0.90–1 = excellent, 0.80–0.90 = good, 0.70–0.80 = fair, 0.60–0.70 = poor, 0.50–0.60 = fail [12].Other cutoffs were considered also. A *p*-value < 0.05 was taken to indicate statistical significance.

## 3. Results

We included in the study 145 patients with COVID-19, 219 patients with different chronic respiratory diseases (interstitial lung disease, OSA-COPD overlap syndrome, bronchiectasis,), and 161 healthy individuals in a control group. Their age, gender, and blood characteristics are shown in Table 1. After adjusting for type one error, NLR was significantly higher in both groups with respiratory disease compared with the control group, but with no statistically significant difference between the two groups. On the other hand, PLR was significantly different between groups. The chronic respiratory disease group was significantly lower than the control group, in which PLR was considerably lower than the COVID-19 group; statistical significance was maintained even after adjusting for type one error. White blood cells(WBC) and neutrophils were higher in the chronic respiratory disease group than in the other two groups, which were not significantly different. Lymphocytes were lower in the COVID-19 group compared with the other two groups, which were not significantly different. Platelets, CRP and ESR were statistically significantly higher in the chronic respiratory disease group compared with the control group, but we did not find a significant difference between the COVID-19 group and the controls, or between the COVID-19 group and chronic respiratory disease group. The chronic respiratory disease group was significantly older than the COVID-19 group, which was considerably older than the control group. Gender was quite different; more male patients were in the chronic respiratory disease group compared with the COVID-19 group, in which there were statistically significantly more males than in the control group.

PLR was found to be significant as a diagnostic test between the COVID-19 group and the other groups. We compared the COVID-19 group versus control group (Figure 1a), COVID-19 group versus chronic respiratory disease group (Figure 1b), and when we summed up the chronic respiratory disease group and control, also (Figure 1c). NLR was poor when discriminating between the COVID-19 and control group, but with statistic significance (Figure 1a); it failed to discriminate in the case of the COVID-19 group versus chronic respiratory disease group (AUC = 0.501, *p* = 0.975) (Figure 1b) and also when we summed the chronic respiratory disease group and control (AUC = 0.553, *p* = 0.060) (Figure 1c). PLR was poor when discriminating between the COVID-19 and control group, but with statistic significance (Figure 1a), poor also at discriminating in the case of the COVID-19 group versus chronic respiratory disease group (Figure 1b), and also when we summed the chronic respiratory disease group and control (Figure 1c).

The optimum cutoff for PLR was found to be 182.48 between the COVID-19 group and control group, 144.95 between the COVID-19 group and chronic respiratory disease group and 157.23 between the COVID-19 group and the other two groups (Table 2), and for NLR, was 3.02 between the COVID-19 group and control group.

### PLR and NLR as Diagnostic Test for Chronic Disease: OSA-COPD Overlap, Interstitial Lung Diseases and Bronchiectasis

In our study, 85 patients had sleep apnea or COPD, 36 patients had interstitial lung diseases, and 98 patients had bronchiectasis. Their age, gender and blood characteristics are shown in Table 3. NLR was not found to be significantly different between groups with different chronic diseases after adjusting for type one error. On the other hand, PLR was significantly different between groups. The OSA-COPD group PLR was significantly lower than in the other two groups, which were not significantly different from the control group. Lymphocytes were higher in the OSA-COPD group compared with the interstitial lung disease group. Conversely, neutrophils were lower in the OSA-COPD group compared with the interstitial lung disease group. Platelets were statistically significantly higher in the bronchiectasis group compared with the OSA-COPD group. The OSA-COPD group had significantly lower ESR than the other two groups, which were not significantly different. Age and CRP were not significantly different between groups.

PLR was found to be poor, but statistically significant as a diagnostic test when we compared the control group with the OSA-COPD group (coded with 1) (Figure 2a); but failed and was not statistically significant when we compared the control group with the interstitial lung disease group (AUC = 0.556, *p* = 0.298) (Figure 2c), or with the bronchiectasis group (AUC = 0.483, *p* = 0.639) (Figure 2e). When we compared the diseases, PLR was found to be poor also and statistically significant in the following cases: the OSA-COPD group (coded with 0) versus interstitial lung disease and bronchiectasis groups (Figure 2b); and the interstitial lung diseases group (coded with 1) versus OSA-COPD and bronchiectasis groups (Figure 2d); but failed and was not statistically significant for the bronchiectasis (coded with 1) group versus OSA-COPD and interstitial lung disease groups (AUC = 0.564, *p* = 0.103) (Figure 2e).

NLR was found to be poor, but statistically significant as a diagnostic test when we compared the control group with all three groups (separately): the OSA-COPD group (coded with 1) (Figure 2a); interstitial lung disease group (Figure 2c); and bronchiectasis group (Figure 2e). When we compared the diseases, NLR failed and was not statistically significant in any of the comparisons: the OSA-COPD group (coded with 0) versus interstitial lung disease and bronchiectasis groups (AUC = 0.507, *p* = 0.859) (Figure 2b); interstitial lung disease group (coded with 1) versus OSA-COPD and bronchiectasis groups (AUC = 0.534, *p* = 0.516) (Figure 2d); and bronchiectasis (coded with 1) group versus OSA-COPD and interstitial lung disease groups (AUC = 0.488, *p* = 0.756) (Figure 2e).

The optimum cutoff for PLR was found to be 114.9 between the OSA-COPD group and control group, 118.38 between the OSA-COPD and interstitial lung disease and bronchiectasis groups, and 101.74 between the interstitial lung disease group (coded with 1) versus OSA-COPD and bronchiectasis groups (Table 4).

The optimum cutoff for NLR was found to be 2.19 between the OSA-COPD group and control group, 3.31 between interstitial lung disease group and control group, and 2.31 between bronchiectasis group and control group (Table 4).

## 4. Discussion

Our study aimed to analyze the diagnostic value of two widely available inflammatory markers, the neutrophil-to-lymphocyte ratio and platelet-to-lymphocyte ratio, in different respiratory diseases: COVID-19, bronchiectasis, OSA-COPD, and interstitial lung disease compared with healthy persons. NLR was found to be significantly higher in both acute (COVID-19) and chronic respiratory disease (OSA-COPD, interstitial lung disease and bronchiectasis) groups when compared with the control group, but with no statistically significant difference between the two groups. NLR was poor when discriminating between COVID-19 and the control group but statistically significant when used to discriminate between COVID-19 and the other chronic respiratory disease group (AUC = 0.501, *p* = 0.975). The optimum cutoff for NLR was 3.02 between the COVID-19 group and the control group. NLR was found to be poor but statistically significant as a diagnostic test when we compared the control group with all three disease groups (separately): OSA-COPD group, interstitial lung disease group, and bronchiectasis group. When we compared the diseases, NLR failed and was not statistically significant in any of the comparisons. The optimum cutoff for NLR was found to be 2.19 between the OSA-COPD group and control group, 3.31 between the interstitial lung disease group and control group, and 2.31 between the bronchiectasis group and control group. There is a large amount of available data evaluating respiratory conditions such as COPD or bronchiectasis in an acute and stable phase, reporting higher values in the first category [13,14,15,16]. As NLR is a very affordable and reproducible marker, the literature is abundant with studies using NLR from sepsis to cancer to restless leg syndrome [17]. However, as it seems to be an indicator of many if not all diseases, we might conclude that it is not a reliable indicator of any disease, as it cannot be a “magical assay for every condition” [18]. We would emphasize that its use is very dependent of clinical context. For example, as mentioned before, NLR was higher in acute exacerbation of COPD and/or bronchiectasis when compared with stable conditions and, of course, when compared with the control group. Several studies that evaluated the ability of NLR to detect bacteremia showed a poor prognostic [19,20,21,22,23]. At a cutoff of ~10, for instance, NLR has a sensitivity of 72% and specificity of 60% for the diagnosis of bacteremia [19]. Nevertheless, its performance in this situation is superior to that of the white blood cell count [18]. Another thing that must be taken into consideration is the fact that under physiologic stress, the number of neutrophils increases, while the number of lymphocytes decreases rapidly, in under 6 h [19]. Increased levels of cortisol as well as endogenous catecholamines such as epinephrine are known to increase the neutrophil count while simultaneously decreasing the lymphocyte count [18]. Cytokines and other hormones are also likely to be involved. In conclusion, NLR is not only an indicator of infection or inflammation, but in fact may be increased by any cause of physiologic stress. The prompt response may make NLR a better reflection of acute stress than other laboratory values, making it perhaps more useful in acute rather than chronic conditions. Many patients have severe physiologic stress (with elevated NLR) without bacteremia. Alternatively, some patients with bacteremia tolerate this surprisingly well and are not very ill. In short, it is unrealistic to expect NLR to perform well in this context. This is not a failure of the test itself, but rather represents a failure to apply the test appropriately [18,19,20]. So, patients with inflammatory disorders may tend to present elevated NLR more than in non-inflammatory disorders and NLR in a critically ill patient may be more elevated than in a non-critical patient. As showed before, interpretation of NLR is dependent on clinical context, and there are no standard values; nevertheless, some authors [18] have suggest some values: normal NLR is roughly 1–3. An NLR of 6–9 suggests mild stress, and critically ill patients will often have an NLR of ~9 or higher. This hypothesis does not seem to be sustained by our study, where NLR values seemed to be similar in the COVID-19 group (acute condition) and chronic respiratory disease group (OSA-COPD, bronchiectasis, interstitial lung disease and pleural effusions). The explanation might be the fact that our COVID-19 population was quite heterogenousas in Romania, unlike other countries, hospitalization was compulsory for all COVID-19 patients, regardless of the severity of the disease. NLR was not able to differentiate between the chronic conditions. As NLR is influenced by steroids and acute respiratory conditions such as COPD or ILD, and in some situations COVID-19 patients might receive systemic steroids, we may ask whether this does not contribute to the elevated values.

PLR, on the other hand, was significantly different between groups.PLR was found to be significant as a diagnostic test between the COVID-19 group and the other groups. PLR was poor when used to discriminate between COVID-19 and the control group with statistical significance and also poor when used to discriminate between the COVID-19 group and chronic respiratory disease group. The optimum cutoff for PLR was found to be 182.48 between the COVID-19 group and control group, 144.95 between the COVID-19 group and chronic respiratory disease group, and 157.23 between the COVID-19 group and the other two groups. PLR was found to be poor, but statistically significant, as a diagnostic test when we used it to compare the control group with the OSA-COPD group, bur not with the interstitial lung disease group (AUC = 0.556, *p* = 0.298) or bronchiectasis group (AUC = 0.483, *p* = 0.639).

When we compared diseases, PLR was found to also be poor, but statistically significant in the following cases: the OSA-COPD group versus interstitial lung disease and bronchiectasis groups; and interstitial lung disease group versus OSA-COPD and bronchiectasis groups. However, it failed and was not statistically significant for the bronchiectasis (coded with 1) group versus OSA-COPD and interstitial lung disease groups (AUC = 0.564, *p* = 0.103) (Figure 2e).

The optimum cutoff for PLR was found to be 114.9 between the OSA-COPD group and control group, 118.38 between OSA-COPD and interstitial lung disease and bronchiectasis groups, and 101.74 between interstitial lung disease group (coded with 1) versus OSA-COPD and bronchiectasis groups.

PLR in the chronic respiratory disease groups was significantly lower than that of the control group, in which PLR was significantly lower than that of the COVID-19 group. The statistical significance was maintained even after adjusting for type one error. PLR was found to be significant as a diagnostic test between the COVID-19 group and the other groups. The optimum cutoff for PLR was found to be 182.48 between the COVID-19 group and control group, 144.95 between the COVID-19 group and chronic respiratory disease group, and 157.23 between the COVID-19 group and the other two groups, but the test accuracy was poorAs a discriminative marker between the COVID-19 group and control group, NLR had statistical significance but was poor as a diagnostic test. NLR was not found to be significantly different between groups of different chronic diseases after adjusting for type one error. On the other hand, PLR was significantly different between groups. The OSA-COPD group PLR was significantly lower than that of the other two groups, which were not significantly different from the control group. PLR was found to be significant in the following cases: the OSA-COPD group versus interstitial lung disease and bronchiectasis groups, and interstitial lung disease group versus OSA-COPD and bronchiectasis groups. NLR was found to be significant as a diagnostic test but poor when we compared the control group with all three groups (separately), but not when we compared among respiratory diseases groups. One can consider a cutoff with high sensitivity if they want to use NLR or PLR as screening tests. In the case of COVID-19, they can be useful as screening tests. Although both NLR and PLR can be very useful screening tools in COVID-19, it is highly unlikely that they will be used as they cannot prevent the disease, which is treated only when symptomatic. When these hematological changes appear, most likely the disease has already started. One can consider a cutoff with high specificity if they want to use NLR or PLR as a precision diagnostic test. The platelet-to-lymphocyte ratio appeared to be a more reliable diagnostic factor than NLR in the present study. PLR is used more as a diagnostic tool in cardiovascular disease (myocardial infarction and vascular diseases). Even in patients with COPD-OSA, it seems to have better discriminative value than NLR, which suggests that hypoxemia, more that inflammation, has a certain influence on the PLR value. This might explain its discriminative value among patients with conditions accompanied by hypoxemia, such as interstitial lung disease and OSA-COPD [10,11].

Our study had some limitations. It was a single-center, retrospective study. Sample size was small for the interstitial lung disease group. The groups were inhomogeneous in terms of number of participants, age, and gender. We had an acute condition as represented by a viral infection (COVID-19) and three chronic conditions (among many others). We did not have acute conditions correspondent with the above-mentioned chronic pathologies (COPD exacerbation, bronchiectasis exacerbation). Study population was quite heterogenous and not matched in age and gender, due first to differences in prevalence of the diseases among different age categories. In the COPD-OSA overlap, bronchiectasis cases were prevalent, whereas interstitial lung diseases were not. The conclusions could be influenced by the differences in age and gender between the control group and the other groups because many studies have found a positive correlation between NLR and age [23]. For comparisons with a control group, age and gender can be confounding factors and careful conclusions should be considered. As another limitation, disease severity was not evaluated in the present study. As we do know that NLR values increase with the severity of some diseases, it would be interesting to perform a subgroup analysis. We also had only one determination of both NLR and PLR. Repetitive determinations at different moments in time would be more interesting. Multiple determinations might have higher prognostic value than an isolated one.

However, ours is among the few studies that evaluated these accessible and very practical inflammatory markers for so many respiratory diseases. Although it was retrospective in nature, it was s a real-life study from a major hospital in Romania. Despite its limitations, we believe that some conclusions can be drawn:NLR was found to be poor as a diagnostic test when we compared healthy persons with patients with chronic respiratory diseases, but not when we compared the diseases.PLR seems to be a more reliable marker in differentiating between the evaluated chronic diseases, but as a diagnostic test, it remains poor.

## 5. Conclusions

The PLR has the potential to be a precision diagnostic tool for COVID-19 and a screening tool for chronic disease. However, diagnostic phase III and IV studies are needed to further evaluate its benefits and clinical relevance. Additionally, we did not manage to find any cutoff value for diagnosis.

## Figures and Tables

**Figure 1 diagnostics-12-00081-f001:**
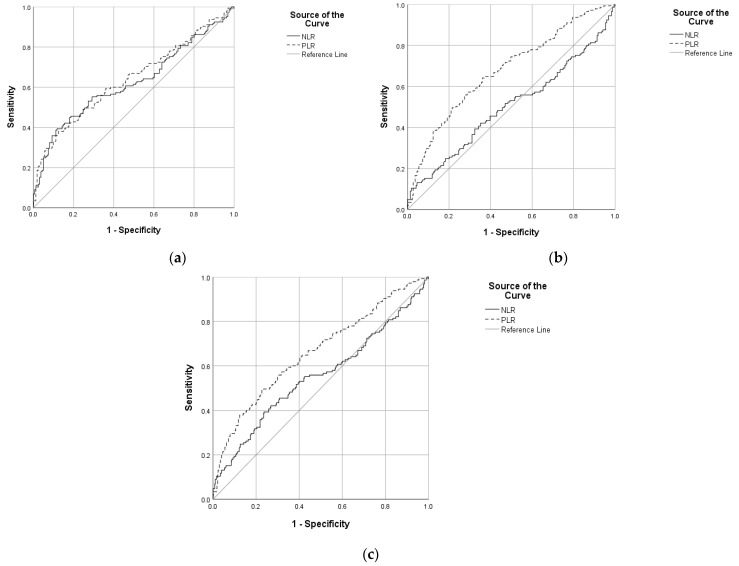
ROC curve with PLR and NLR (**a**) comparing COVID-19 group versus control group; (**b**) comparing COVID-19 group versus chronic respiratory disease group; (**c**) comparing COVID-19 group versus control group and chronic respiratory disease group.

**Figure 2 diagnostics-12-00081-f002:**
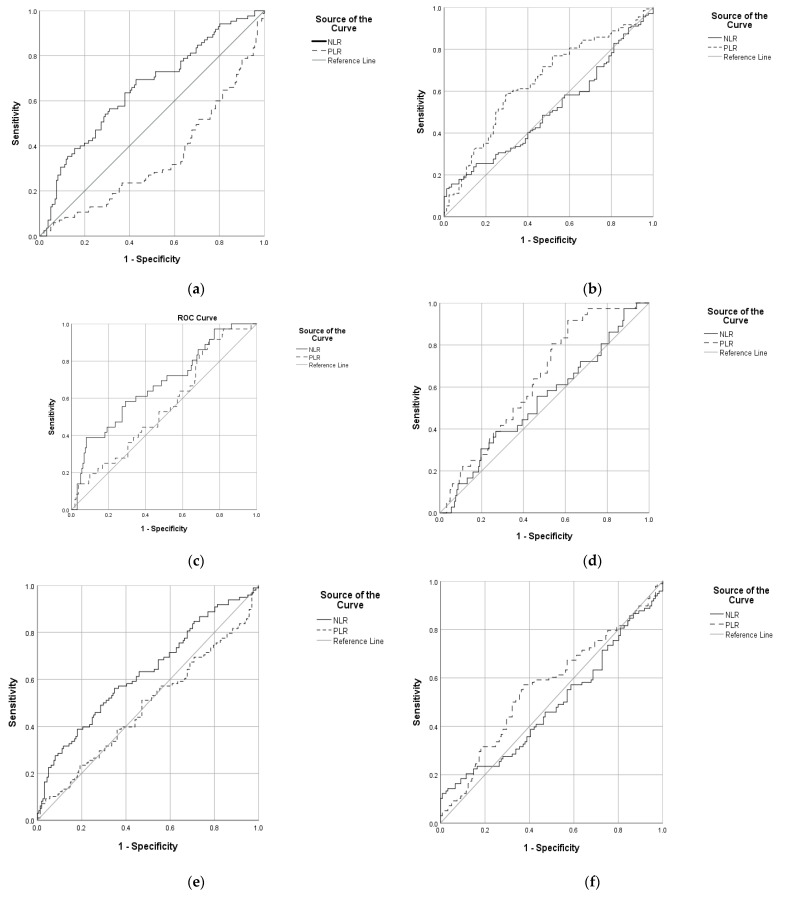
ROC curve with PLR and NLR (**a**) comparing OSA-COPD group (coded with 0) versus control group; (**b**) comparing OSA-COPD group (coded with 0) versus interstitial lung disease and bronchiectasis groups; (**c**) comparing interstitial lung disease group (coded with 1) versus control group; (**d**) comparing interstitial lung disease group (coded with 1) versus OSA-COPD and bronchiectasis groups; (**e**) comparing bronchiectasis (coded with 1) group versus control group; (**f**) comparing bronchiectasis (coded with 1) group versus OSA-COPD and interstitial lung disease groups.

**Table 1 diagnostics-12-00081-t001:** The characteristics of the chronic disease group compared with the control group and COVID-19 group (*n* = 145, data not presented, as they have already been published [3]).

	COVID-19 Group (*n* = 145)	Other Disease Group (*n* = 219)	Control Group (*n* = 161)	*p*
AGE	46 (33.5, 57) ^a,b^	62 (52.5, 68) ^c^	40 (29, 48)	<0.001 **
Males, number (%)	69 (47.6) ^a,b^	138 (65.1) ^c^	33 (20.5)	<0.001 **
NLR	2.56 (1.72, 3.79) ^b^	2.48 (1.85, 3.49) ^c^	2.03 (1.59, 2.59)	<0.001 **
PLR	151.85 (112.86, 211.59) ^a,b^	114.1 (92.31, 150.13)	125.37 (101.78, 156.71)	<0.001 **
White blood cells	5.95 (4.89, 8.05) ^a^	7.77 (6.25, 9.45) ^c^	6.3 (5.54, 7.78)	<0.001 **
Lymphocytes (103/μL)	1.56 (1.2, 2.03) ^a,b^	1.93 (1.52, 2.35)	1.96 (1.62, 2.34)	<0.001 *
Neutrophils (103/μL)	4.01 (2.94, 5.5) ^a^	4.94 (3.94, 6.34) ^c^	3.96 (3.19, 4.94)	<0.001 **
Thrombocytes (X103)	249 (183, 299)	231 (189, 273.5) ^c^	247 (222, 279)	0.016 **
CRP	3.25 (0.99, 18.7)	4.9 (2.2, 11.8) ^c^	2.55 (1.5, 4.4)	0.006 **
ESR	12 (5.5, 30)	15 (7, 28) ^c^	11 (6, 15)	0.003 **

* *p* from Anova test between three groups: the chronic disease group, the control group, and the COVID-19 group [3] (data not shown); ** *p* from Kruskal–Wallis test between three groups: the chronic disease group the control group and the COVID-19 group [3] NLR—neutrophil-to-lymphocyte ratio; PLR—platelet-to-lymphocyte ratio; CRP—C-reactive protein; ESR—erythrocyte sedimentation rate; ^a^—adjusted *p* < 0.05 for COVID-19 group compared with other disease group; ^b^—adjusted *p* < 0.05 for COVID-19 group compared with control group; ^c^—adjusted *p* < 0.05 for other disease group compare with control group.

**Table 2 diagnostics-12-00081-t002:** Performance of PLR and NLR in the case when statistically significant difference in ROC curve was reached.

		Cutoff	Sensitivity	Specificity
PLR	COVID-19 group (codified with 1) vs. control—AUC = 0.640, *p* < 0.001	90.78	0.90	0.16
		112.52	0.75	0.35
		**182.48**	**0.38**	**0.88**
		190.48	0.33	0.90
	COVID-19 group (codified with 1) vs. chronic respiratory disease group—AUC = 0.677, *p* < 0.001	90.5	0.90	0.24
		112.68	0.75	0.48
		**144.95**	**0.56**	**0.73**
		195.84	0.30	0.90
	COVID-19 group (codified with 1) vs. control and chronic respiratory disease group—AUC = 0.662, *p* < 0.001	90.78	0.90	0.21
		112.68	0.75	0.43
		**157.23**	**0.49**	**0.77**
		193.55	0.30	0.90
NLR	COVID-19 group (codified with 1) vs. control—AUC = 0.624, *p* < 0.001	1.35	0.90	0.12
		1.72	0.75	0.32
		**3.02**	**0.39**	**0.88**
		3.15	0.36	0.90

The cutoffs with maximal sensitivity and specificity are marked in bold; NLR—neutrophil-to-lymphocyte ratio; PLR–platelet-to-lymphocyte ratio; AUC—area under the curve.

**Table 3 diagnostics-12-00081-t003:** The characteristics of the chronic disease groups.

	OSA-COPD Overlap (*n* = 85)	Interstitial Lung Diseases (*n* = 36)	Bronchiectasis (*n* = 98)	*p*
AGE	61 (51, 65)	64 (57, 69.5)	62 (54, 74)	0.177
Gender, number (%)	66 (84.6) ^a,b^	20 (55.6)	52 (53.1)	<0.001
NLR	2.5 (1.87, 3.33)	2.54 (1.82, 3.76)	2.38 (1.81, 3.46)	0.808
PLR	103.93 (84.85, 129.14) ^a,b^	129.8 (109.62, 165.94)	129.18 (95.18, 162.43)	0.001
Lymphocytes (103/μL)	2.04 (1.68, 2.38) ^a^	1.71 (1.31, 2.06)	1.95 (1.47, 2.38)	0.012
Neutrophils (103/μL)	5.41 ± 1.77 ^a^	4.61 ± 1.79	5.57 ± 2.82	0.037
Platelets (X103)	224 (189, 256) ^b^	229.5 (175.5, 274)	241.5 (202, 295)	0.043
ESR	8 (5, 19) ^a,b^	18.5 (12.5, 31)	20 (11, 34.5)	<0.001
CRP	4.8 (2.4, 11.7)	3.45 (1.85, 10.15)	6.05 (1.95, 16.65)	0.379

^a^—adjusted *p* < 0.05 for OSA-COPD group compared with interstitial lung disease group; ^b^—adjusted *p* < 0.05 for OSA-COPD group compared with bronchiectasis group.

**Table 4 diagnostics-12-00081-t004:** Performance of PLR and NLR when we compare chronic respiratory disease.

		Cutoff	Sensitivity	Specificity
PLR	OSA-COPD group (codified as 0) vs. control—AUC = 0.656, *p* < 0.001	84.41	0.90	0.17
		101.82	0.75	0.48
		114.9	0.63	0.68
		172.98	0.17	0.90
	OSA-COPD group (coded with 0) versus interstitial lung disease and bronchiectasis groups—AUC = 0.645, *p* < 0.001	75.14	0.90	0.24
		102.71	0.75	0.48
		118.38	0.58	0.71
		173.78	0.19	0.90
	Interstitial lung disease group (coded with 1) versus OSA-COPD and bronchiectasis groups—AUC = 0.636, *p* = 0.010	101.74	0.92	0.39
		109.62	0.75	0.47
		148.90	0.36	0.75
		189.88	0.19	0.90
NLR	OSA-COPD group (codified with 1) vs. control—AUC = 0.652, *p* < 0.001	1.56	0.90	0.22
		2.19	0.69	0.57
		2.59	0.47	0.75
		3.15	0.31	0.90
	Interstitial lung disease group (codified with 1) vs. control—AUC = 0.672, *p* = 0.001	1.60	0.90	0.25
		1.84	0.75	0.37
		2.63	0.47	0.75
		**3.31**	**0.39**	**0.92**
	Bronchiectasis disease group (codified with 1) vs. control—AUC = 0.672, *p* = 0.001	1.49	0.90	0.20
		1.80	0.75	0.36
		**2.31**	**0.56**	**0.64**
		3.15	0.29	0.90

The cutoffs with maximal sensitivity and specificity are marked in bold; NLR—neutrophil-to-lymphocyte ratio; PLR—platelet-to-lymphocyte; AUC—area under the curve.

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
