# Peer review of "Diagnostic Value of the Neutrophil-to-Lymphocyte Ratio (NLR) and Platelet-to-Lymphocyte Ratio (PLR) in Various Respiratory Diseases: A Retrospective Analysis"

_diagnostics, 2021, doi:10.3390/diagnostics12010081_

Round 1

Reviewer 1 Report

Overall the manuscript reads well. However there are some missing pieces of information as follows-

1) Exactly how were the control group recruited? It appears that they may have been screened for respiratory disease but this needs to be made clearer. Is there any chance that any of the control group had COVID or was this ruled out by them having to have a negative COVID test?

2) The discussion section lacks a paragraph on what next. Does this study need to be repeated in a subset of your patients with COVID and/or respiratory disease and if so which subset? Would serial measurements be of greater use than a one off measurement? How is this information best used in combination with other laboratory parameters?

Author Response

  1. Exactly how were the control group recruited? It appears that they may have been screened for respiratory disease but this needs to be made more explicit. Is there any chance that any of the control groups had COVID, or was this ruled out by having a negative COVID test?

The control group is the healthy medical staff that presented for annual evaluation. They were clinically examined and had blood tests performed. The patients were evaluated before covid-19 pandemic started so they did not had covid-19

2) The discussion section lacks a paragraph on what next. Does this study need to be repeated in a subset of your patients with COVID and/or respiratory disease and if so which subset? Would serial measurements be of greater use than a one off size? How is this information best used in combination with other laboratory parameters?

These are great questions, and we should address them in our discussion section. Thank you very much

Reviewer 2 Report

  The present study tries to find out if a correlation exists between PLR and NLR levels in healthy persons and PLR and NLR levels in patients with various respiratory diseases. Besides, it also aims to find out whether PLR and NLR levels can be used to differentiate acute respiratory diseases (COVID-19) from chronic respiratory diseases (OSA-COPD, interstitial lung diseases and bronchiectasis). The article covers a trending topic as it targets respiratory diseases which are of high interest specially after the COVID-19 outbreak. The authors do a good job in study design and a good statistical work have been implemented. The conclusions are reasonablyy drawn according to the obtained results.

The study however brings some limitations:

  • I am a little bit concerned about the considerable higher median age of the COVID-19 group and specially the other disease group respect to the control group. Since many studies have found a positive correlation between NLR and age, e.g. doi: 1002/jcla.21791, drawing careful conclusions is advised. This is one of the reasons that in reference 3 the median age in the control group is exactly matched to the median of the COVID-19 group.
  • The difference in the percentage of sex genre is evident among the different groups: 65% male percentage in the other disease group versus only 20% male percentage in the control group.
  • The interstitial lung diseases group is only formed by 36 patients, which represents a low dataset.

Please provide an explanation about these possible biases in the study.

A rough guide for classifying the accuracy of a diagnostic test is the traditional academic point system:

  • .90-1 = excellent (A)
  • .80-.90 = good (B)
  • .70-.80 = fair (C)
  • .60-.70 = poor (D)
  • .50-.60 = fail (F)

Explain the relevance of the obtained AUC values with the above classification and discuss the feasibility of any implementation at a healthcare level.

The meaning of several sentences is not clear. Please modify them. Examples:

  • “AUC was compared with 50% of the square area and was statistically significant the AUC was significant different than it, which mean that the considered parameter had the power to discriminate between the tested groups”
  • “An age-matched group of the same pathologies (COPD, bronchiectasis, interstitial lung disease) but in exacerbation would have been useful”.

Not using a reference manager like Endnote, Zotero, Medeley is risky. Please provide complete reference for the manuscripts. Examples:

  • Man MA, Rajnoveanu R-M, Motoc NS, Bondor CI, Chis AF, Lesan A, et al. (2021) Neutrophil-to-lymphocyte ratio, platelets-to-lymphocyte ratio, and eosinophils correlation with high-resolution computer tomography severity score in COVID-19 patients. PLoS ONE 16(6): The year of publication is missing
  • Sun, Y.; Chen, C.; Zhang, X.; Weng, X.; Sheng, A.; Zhu, Y.; Chen, S.; Zheng, X.; Lu, C. High Neutrophil-to-Lymphocyte Ratio Is an Early Predictor of Bronchopulmonary Dysplasia. Front. Pediatr.2019, 7. The pages are missing

There are many typos and grammatical errors. Please modify accordingly. Examples:

  • Page 2. Respiratory instead of respiratory
  • Page 3. Platelets instead of plateletes. We didn’t find instead of we didn’t found.
  • Table 3. Cells instead celles
  • Page 7. Figure 2 instead of Figure 1
  • Page 8. Hours instead of ours.
  • Page 8. Ill patients instead of il patients
  • Page 9. where NLR values seem to similar in the COVID-19
  • Page 10. Platelets-to-lymphocytes ratio. Why not use PLR?
  • Page 10. Which makes us think instead of which makes as think
  • Page 10. Might explain its discriminative instead of might explain it’s discriminative

Introduction should be improved. An extra paragraph should be written to describe the work that has been done and about ROC curves, etc.

Author Response

The study however brings some limitations:

  • I am a little bit concerned about the considerable higher median age of the COVID-19 group and specially the other disease group respect to the control group. Since many studies have found a positive correlation between NLR and age, e.g. doi: 1002/jcla.21791, drawing careful conclusions is advised. This is one of the reasons that in reference 3 the median age in the control group is exactly matched to the median of the COVID-19 group.
  • The difference in the percentage of sex genre is evident among the different groups: 65% male percentage in the other disease group versus only 20% male percentage in the control group.
  • The interstitial lung diseases group is only formed by 36 patients, which represents a low dataset.

Please provide an explanation about these possible biases in the study.

We insert the explanation of these limitations in the discussion part

A rough guide for classifying the accuracy of a diagnostic test is the traditional academic point system:

  • .90-1 = excellent (A)
  • .80-.90 = good (B)
  • .70-.80 = fair (C)
  • .60-.70 = poor (D)
  • .50-.60 = fail (F)

 We modified according with this classification

Explain the relevance of the obtained AUC values with the above classification and discuss the feasibility of any implementation at a healthcare level.

 We explain in term of screening or precision test

The meaning of several sentences is not clear. Please modify them. Examples:

  • “AUC was compared with 50% of the square area and was statistically significant the AUC was significant different than it, which mean that the considered parameter had the power to discriminate between the tested groups”
  • “An age-matched group of the same pathologies (COPD, bronchiectasis, interstitial lung disease) but in exacerbation would have been useful”.

We correct these sentences

Round 2

Reviewer 1 Report

The requisite changes appear to have been made to the manuscript